# Transcranial Photobiomodulation for the Treatment of Children with Autism Spectrum Disorder (ASD): A Retrospective Study

**DOI:** 10.3390/children9050755

**Published:** 2022-05-20

**Authors:** Stefano Pallanti, Michele Di Ponzio, Eleonora Grassi, Gloria Vannini, Gilla Cauli

**Affiliations:** 1Neurodevelopment Division, Istituto di Neuroscienze, 50121 Florence, Italy; ricerca@istitutodineuroscienze.it (M.D.P.); e.grassi@istitutodineuroscienze.it (E.G.); gloria.vannini@virgilio.it (G.V.); 2Department of Psychiatry, Albert Einstein College of Medicine, Bronx, NY 10461, USA; 3Asst Fatebenefratelli Sacco, 20154 Milan, Italy; gilla.cauli@asst-fbf-sacco.it

**Keywords:** autism spectrum disorders, ASD, photobiomodulation, LED, near-infrared, NIR, neuromodulation

## Abstract

Children with Autism Spectrum Disorder (ASD) face several challenges due to deficits in social function and communication along with restricted patterns of behaviors. Often, they also have difficult-to-manage and disruptive behaviors. At the moment, there are no pharmacological treatments for ASD core features. Recently, there has been a growing interest in non-pharmacological interventions for ASD, such as neuromodulation. In this retrospective study, data are reported and analyzed from 21 patients (13 males, 8 females) with ASD, with an average age of 9.1 (range 5–15), who received six months of transcranial photobiomodulation (tPBM) at home using two protocols (alpha and gamma), which, respectively, modulates the alpha and gamma bands. They were evaluated at baseline, after three and six months of treatment using the Childhood Autism Rating Scale (CARS), the Home Situation Questionnaire-ASD (HSQ-ASD), the Autism Parenting Stress Index (APSI), the Montefiore Einstein Rigidity Scale–Revised (MERS–R), the Pittsburgh Sleep Quality Index (PSQI) and the SDAG, to evaluate attention. Findings show that tPBM was associated with a reduction in ASD severity, as shown by a decrease in CARS scores during the intervention (*p* < 0.001). A relevant reduction in noncompliant behavior and in parental stress have been found. Moreover, a reduction in behavioral and cognitive rigidity was reported as well as an improvement in attentional functions and in sleep quality. Limitations were discussed as well as future directions for research.

## 1. Introduction

Autism spectrum disorder (ASD) is a complex neurodevelopmental condition typically characterized by deficits in social and communicative behaviors as well as repetitive patterns of behaviors (APA 2013) [1]. In addition to such core symptoms, several children and adolescents with ASD also present severe behavioral difficulties, including aggression, self-injurious behavior, tantrums, irritability and sleep problems, which usually interfere with their education and development as well as the wellbeing of caregivers (Hill et al., 2014; Soke et al., 2016; Baglioni et al., 2016) [2,3,4]. Moreover, people with ASD showed attentional and executive function deficits (Gargaro et al., 2011; Demetriou et al., 2019) [5,6].

While the cause of autism is uncertain, the most widely accepted explanation is that it is a complex neurodevelopmental disorder characterized by brain network abnormalities. EEG has shown local overconnectivity and long-range underconnectivity, also involving the corpus callosum (Barttfeld et al., 2011) [7]. fMRI studies revealed altered functional connectivity in the default mode network (DMN), a network with a role in interoceptive awareness and mind wandering and which was implicated in social-cognitive deficits of autism (Harikumar et al., 2021; Broyd et al., 2009) [8,9].

The main goal of therapy for children with ASD is the improvement of socio-relational and communication skills. This goal is pursued through a combination of interventions, such as speech therapy, parent training, social skills training and cognitive-behavioral therapy (Chahin et al., 2020) [10]. In the presence of emotional and behavioral dysregulation, a pharmacological approach is often considered (Eissa et al., 2018; Pallanti et al., 2015) [11,12]. Although some medications such as risperidone and aripiprazole have an effect on ASD-related irritability and aggression (DeVane et al., 2019) [13], they also have important side effects including sedation, anticholinergic effects, metabolic alterations, weight gain and involuntary movements (DeVane et al., 2019) [13]. Moreover, they do not target the core features of ASD.

Recently, there has been a growing interest in the potential of non-invasive brain stimulation in neurodevelopmental disorders, thanks to their ability to modulate neuroplasticity and enhance cognitive, behavioral and socio-emotional processes (Finisguerra et al., 2019; Enticott, Pallanti and Hollander, 2018) [14,15].

Among new neuromodulation approaches, transcranial photobiomodulation (tPBM) is characterized by the noninvasive delivery of low-level light, transcranially. Light penetrates the skin and the skull and then is absorbed into the brain tissue by specific chromophores, such as water, oxyhemoglobin (HbO2), deoxyhemoglobin (Hb), myoglobin, melanin, cytochromes, and flavin. The target for light within single neurons is the mitochondria, where tPBM stimulates cytochrome c oxidase. The consequence is that light enhances mitochondrial activity and hence ATP synthesis, leading to an activation of transcription factors associated with increased functional activity (Salehpour et al., 2018; Mitrofanis and Henderson, 2020) [16,17]. Coherently, research has shown that tPBM boosts brain energy metabolism as well as cognition in preclinical (Mochizuki-Oda et al., 2002; Konstantinovic et al., 2013) [18,19] and clinical studies, (Maiello et al., 2019) [20]. tPBM has been effectively applied in post-stroke rehabilitation (Yang et al., 2018) [21], in patients with TBI (Figueiro Longo et al., 2020) [22] and depression (Askalsky and Iosifescu, 2019) [23].

As far as safety is concerned, in a randomized-controlled study, which included about 1000 patients with stroke (Hacke et al., 2014) [24], no significant difference in side effects was reported between active and sham stimulation with tPBM. Other studies reported transient headaches, insomnia, irritable mood and a strange taste in the mouth as the most common side effects (Cassano et al., 2018; 2019) [25,26]. The risk of thermal injury following tPBM is minimal and mostly dependent on the parameters and device used (Caldieraro and Cassano, 2019) [27].

Specifically, concerning ASD in adults, a recent study by Ceranoglu and colleagues (2022) [28] also reported no side effects, with the exception of one out of six patients who developed a transient headache. They suggested beneficial effects of twice-a-week Transcranial Light Emitting Diode (LED) Therapy (TLT), a form of PBM, on core social deficits associated with ASD in adult patients aged 18–55 years, as shown by the reduction in the restricted interests and repetitive behavior subscale of the Social Responsiveness Scale (SRS-2) and on measures of social emotional competence and global functioning, with a good tolerability and adherence rate.

Furthermore, tPBM could also be safely and efficiently used in children and adolescents, considering that several studies used PBM to treat pediatric samples with no reported or minimal side effects (Leisman et al., 2018; Mannu et al., 2019; Salgueiro et al., 2021; Noirrit-Esclassan et al., 2019; Santos et al., 2017) [29,30,31,32,33]. Furthermore, phototherapy, of which PBM is a variant—although the wavelength used in phototherapy is lower than in tPBM—has also been widely adopted in neonates (Faulhaber et al., 2019) [34] and, although some reported side effects, many were transient and mild.

Concerning ASD specifically, Leisman and colleagues (2018) [29] treated children and adolescents with ASD administering low-level light therapy (a form of PBM) to the base of the skull and temporal areas eight times for 5 min and no side effects that necessitate discontinuation of the therapy were reported. All the participants were evaluated with the Aberrant Behavior Checklist (ABC) and there were no dropouts. Results show a decreased irritability after treatment, suggesting the potential of PBM also in treatment of children with ASD.

Based on these preliminary findings, tPBM has been prescripted for home-based treatment of children and adolescents with ASD on the basis of the principle of the good clinical practice. Previously, PBM has been used efficiently and without side effects in other studies for home treatment (Chao, 2019; Gavish and Houreld, 2019) [35,36]. In our study, the type of tPBM employed (Vielight^®^ Neuro Alpha/Gamma stimulator) stimulates the default mode network (DMN) (Vielight, 2020) [37] and not only the temporal lobe, as was the case in the study by Leisman and collegues (2018) [29].

Psychometrical data were collected. Here, they are reported and analyzed retrospectively, with the aim to examine the clinical profile of children and adolescents with ASD before and after treatment with tPBM.

## 2. Materials and Methods

### 2.1. Participants

Clinical data of children and adolescent patients with a diagnosis of ASD according to DSM-5 criteria were extracted from databases containing information on patients of the psychiatric clinic at the Istituto di Neuroscience, Florence (Italy). It is important to mention that the database contains only the data of patients who accepted treatment among all the ones to which was proposed. The diagnosis was confirmed with the Autism Diagnostic Interview–Revised (ADI–R) and a CARS (Schopler et al., 1980) [38] total score of no less than 30. tPBM was added to ongoing behavioral or pharmacological treatments, which were unchanged for at least 1 month at the date of the start of the stimulation and remained unchanged throughout the stimulation period. Demographical data as well as ongoing treatments are reported in Table 1.

After the complete description of the study to participants’ parents, written informed consent was obtained in accordance with the Declaration of Helsinki.

### 2.2. Stimulation

tPBM was delivered using the commercially available Vielight^®^ Neuro Alpha/Gamma brain photo biomodulation stimulator. Two stimulator devices were used: alpha and gamma. The alpha stimulator device delivers 810 nm near infrared light pulsing at 10 Hz via the transcranial LED clusters placed on the Photo-Bio-Modulation helmet. The 10 Hz correlates with alpha brain waves which are produced by the brain during meditation and relaxation states. The gamma stimulator pulses light at 40 Hz light pulsing frequency and delivers 810 nm near infrared light via the transcranial LED clusters placed on the helmet. The frequency of gamma stimulation simulates neural gamma waves which are correlated with increased cognitive activities. Both protocols were used in order to exploit the advantages of both and increase attention, improve sleep and reduce irritability and rigidity.

The device is composed of a wearable headset (see Figure 1) with features microchip-boosted transcranial LED diodes. The tPBM headset consists of four clusters. According to the 10–20 EEG system, the frontal cluster should be positioned over FPz, the posterior cluster over Cz and the two lateral ones over T3 and T4. In this way, the four LEDs deliver the NIR to the subdivisions of the DMN: the medial prefrontal cortex, the precuneus area, and left and right angular gyrus (Vielight, 202) [37]. The intranasal application is positioned in the left or right nostril with the clip on the outside to deliver light to the ventral section of the brain, specifically to the ventromedial PFC. The support pads should fall naturally into place around the ears.

LED diodes emit non-thermal, non-laser light at an intensity that penetrates the scalp, skull, and meninges to a depth of ~40 mm, stimulating cortical brain areas (Jagdeo et al., 2012; Tedford et al., 2015) [39,40] and is powered by three rechargeable NiMH batteries. The posterior transcranial LEDs have a power of 100 milliwatts (mW) and the anterior transcranial has a power of 75 mW. Each posterior transcranial LED has a power density of 100 mW/cm^2^ and the anterior transcranial LED of 75 mW/cm^2^. The beam spot size of each LED is approximately 1 cm^2^. The energy delivered by posterior transcranial LEDs is 60 joules (J) and the anterior transcranial LED delivers 45. The energy density of the posterior transcranial LEDs is 60 J/cm^2^ and 45 J/cm^2^ for the anterior transcranial LED. The hamma and alpha stimulator devices delivered 240 J during a 20-minute treatment session. For both gamma and alpha stimulations, an intranasal neurostimulator was used to simultaneously stimulate ventral brain areas. The intranasal neurostimulator has an 810 nm wavelength near infrared light LED that delivers NIR through the nasal channel. The intranasal LED has a power of 25 mW and a power density of LED of 25 mW/cm^2^. The energy delivered by the intranasal LED is 15 Joules; the energy density is 15 J/cm^2^. Light parameters are summarized in Table 2.

The stimulation is painless, non-invasive, and well-tolerated. The PBM devices used in this study are considered to be non-regulated, ‘‘low risk general wellness products,’ according to the “General Wellness: Policy for Low Risk” published by the Food and Drug Administration in September 2019 [41].

### 2.3. Procedure of Administration

Patients received tPBM treatments at home for 5 days a week, for 6 months (from November 2020 to April 2021). Parents were trained in how to position the tPBM device and administer the protocols. For the first at-home session, parents were asked to contact the staff via videocall so that could be possible to control the correct administration of the protocols and, if necessary, correct possible mistakes. Parents were retrained when necessary and they were contacted every week by the staff to assess for adverse events. An alpha and a gamma protocol were administered each day, one in the morning and one in the evening. Each session had a duration of 20 min, during which children were involved in stimulating activities (such drawing, coloring, reading, playing games, or doing homework).

### 2.4. Baseline and Follow-Up Assessments and Outcome Measures

Baseline assessments were performed before the first tPBM session and repeated after three and six months. Safety and tolerability were monitored by assessing adverse events and vital signs weekly.

The primary outcome was the change from baseline to 3- and 6-month in the Childhood Autism Rating Scale (CARS) (Schopler et al., 1980; 1988) [38,42]. The CARS consists of 14 domains assessing behaviors associated with autism, with a 15th domain rating the general impression of autism. Total score ranges from 15 to 60, with scores below 30 indicating the absence of autism, a score ranging between 30 and 36.5 indicating mild-to-moderate autism, and scores higher than 37 indicating severe autism (Schopler et al., 1988) [42].

Secondary outcomes were measured using the Home Situation Questionnaire-ASD (HSQ-ASD), the Autism Parenting Stress Index (APSI), the SDAG (Scala per i Disturbi di Attenzione/Iperattività per Genitori (ADHD rating scale for Parents)), the Montefiore Einstein Rigidity Scale–Revised (MERS–R) and the Pittsburgh Sleep Quality Index (PSQI). HSQ-ASD (Chowdhury et al., 2015) [43] is a 24-item parent-rated measure of noncompliant behavior in children with ASD. The scale yields per-item mean scores of 0 to 9 (higher is worse). APSI (Silva and Shalock, 2012) [44] is a 13-item parent-rated measure, which assesses parenting stress in three categories: core social disability, difficult-to-manage behavior, and physical issues. SDAG was completed by the parents. Nine items (marked with odd numbers) explore Inattention (subscale In), and nine items (marked with even numbers) explore Hyperactivity/Impulsivity (subscale H/I). The frequency and intensity of the 18 ADHD symptoms are rated on a 4-point Likert scale from 0 to 3 (0, never, 1, sometimes, 2, often, 3, very often). 

MERS–R measures three domains: behavioral rigidity, cognitive rigidity and protest domain. Behaviors are rated on a scale from 0 to 4. All three domains consist of four items. Regarding behavioral rigidity and cognitive rigidity, the items are: 1, Time spent engaging in behavior; 2., Interference due to behavior; 3, Distress due to disruption of behavior; 4, Degree of control. Regarding the protest domain, the items are: 1, Time spent protesting; 2, Interference due to protest; 3, Severity of protest; 4, Effort for redirection.

PSQI (Buysse et al., 1989) [45] is a standardized self-administered questionnaire, that in this case was completed by parents. It aims to assess sleep problems and its quality.

### 2.5. Statistical Analysis

The baseline demographic and clinical characteristics of the sample were tabulated with descriptive statistics. Parametric and non-parametric tests were used according to variables’ distribution to analyze changes in scores over time. For all statistical analyses, the alpha level of significance was set at 0.05. All the statistical analyses were performed using the statistical programming language R (version 4.0.5) (R Core Team. R: A Language and Environment for Statistical Computing. Vienna: R Foundation for Statistical Computing (2021)).

## 3. Results

The study included 21 patients (13 males, 8 females). The average age was 9.1 (range 5–15).

As CARS scores, MERS scores and Inattention subscale of SDAG scores were normally distributed (verified through the Shapiro–Wilk test), one-way repeated measures ANOVA was used to determine whether there were differences in scores during time. CARS results (see Figure 2) showed that they were statistically significantly different at the different time points during tPBM intervention (F (2,40) = 137.143, *p* < 0.001, η^2^g = 0.02). Pairwise comparisons using the Bonferroni correction showed that there was a decrease in CARS score from pre-intervention to three months (*p* < 0.001) and from pre-intervention to six months (*p* < 0.001) as well as from three to six months (*p* < 0.001).

MERS scores (see Figure 3) showed that they were statistically significantly different at the different time points during tPBM intervention (F (2,40) = 116.308, *p* < 0.001, η^2^g = 0.55). Pairwise comparisons using the Bonferroni correction showed that there was a decrease in MERS score from pre-intervention to three months (*p* < 0.001) and from pre-intervention to six months (*p* < 0.001) but not from three to six months (*p* > 0.05).

SDAG scores (see Figure 4) showed that they were statistically significantly different at the different time points during tPBM intervention (F (2,40) = 39.966, *p* < 0.001, η^2^g = 0.574). Pairwise comparisons using the Bonferroni correction showed that there was a decrease in SDAG scores from pre-intervention to three months (*p* < 0.001) and from pre-intervention to six months (*p* < 0.001) as well as from three to six months (*p* < 0.001).

As HSQ-ASD, APSI and PSQI scores were not normally distributed, a Friedman test was run to determine whether there were differences in scores during treatment. HSQ-ASD scores (see Figure 5) were statistically significantly different at the different time points during t-PMB intervention (χ2(2) = 38, *p* = < 0.001, W = 0.905). Post hoc analysis revealed statistically significant differences in the scores between baseline and mid- (*p* < 0.001), and post-treatment (*p* < 0.001), and also between mid- and post-treatment (*p* < 0.01).

A statistically significant difference has also been found in APSI scores (see Figure 4) during intervention (χ2(2) = 39.4, *p* ≤ 0.001, W = 0.938). In this case, post hoc analysis revealed statistically significant differences in the scores between baseline and mid- (*p* < 0.001) and post-treatment (*p* < 0.001), but not between mid- and post-treatment (*p* > 0.05).

PSQI scores (see Figure 6) were statistically significantly different at the different time points during t-PMB intervention (χ2(2) = 18.9, *p* ≤ 0.001, W = 0.451). Post hoc analysis revealed statistically significant differences in the scores between baseline and mid- (*p* < 0.01), and post-treatment (*p* < 0.01), but not between mid- and post-treatment.

As far as safety is concerned, in our study, tPBM sessions were well tolerated: we had no dropouts, and no patient experienced seizures or syncope, neurological complications, or other major adverse effects. Occasional headache has been reported by two patients (9.5% of patients), but the intensity did not require tPBM discontinuation.

## 4. Discussion

The main result of this retrospective study is the reduction in ASD severity as shown by the decrease in CARS scores after the intervention. Then, a reduction in cognitive and behavioral rigidity, measured through the MERS–R, and an increase in sleep quality, measured through the PSQI, were observed. Importantly, attention improved too, as shown by the reduction in the scores of the inattention subscale of the SDAG. A relevant reduction in noncompliant behavior as measured by HSQ-ASD has been also found.

It is noteworthy to mention that these improvements, which have a great impact on patients’ lives, allow for a decrease in parental stress, as measured through the APSI, a result that could lay the foundation for a quieter, more effective growth environment.

It is likely, even if not demonstrable at the moment, that these effects are a consequence of the combination of the two protocols, alpha and gamma. Indeed, increased relaxation, a feature of the alpha protocol, would allow for a reduction in rigidity and sleep improvement, while enhanced cognition effect, an effect of the gamma protocol, would account for improvements in attentional functions.

The positive effects of tPBM are therefore in line with those reported by Ceranoglu and colleagues (2022) [28] on adults with ASD, despite the different protocols, devices and evaluation tools used. Moreover, results are consistent with the study by Leisman and colleagues (2018) [29]. Regardless, in our study tPBM stimulates not only the temporal lobe but the DMN, and this can explain the effects reported here that also concern rigidity and attention. It is significant to report that the device used in our study and the one used in the study by Leisman (Leisman et al., 2018) [28] are different, although they are all forms of photobiomodulation. In this study, the device uses LEDs, while in the other one laser light is used. They differ mainly in light emission. Laser is characterized by coherence while LED is characterized by non-coherence. (Heiskanen and Hamblin, 2018) [46] Nevertheless, the basic working principle is the same in both and their effects are similar (Brochetti et al., 2017) [47]. For our study, we have decided to employ LEDs for the ease of using them at home and because of the lower safety concern associated with their use (Vielight Device emits non-thermal light). Moreover, LEDs can irradiate larger areas of tissue, which is particularly suitable for brain stimulation and specifically for frontal regions stimulation (Salehpour et al., 2018) [16]. In addition, an intranasal light delivery method has also been employed in this study in order to overcome the penetration limitation of LEDs in comparison to laser.

Results regarding improved attention are consistent with a previous study (Jahan et al., 2019) [48], which reported that light irradiation with 850 nm LED source on the right prefrontal cortex improved attentional performance. Despite the significant results, further studies are needed to confirm attention improvements through an evoked potential evaluation.

Improvement in autism severity, which eventually corresponds to an improved cohabitation with their relatives, as a consequence of tPBM, as reported here, could also be explained due to the potential effect over electrophysiological oscillations. Indeed, EEG power abnormalities in autism have been reported (Wang et al., 2013) [49] and recently, tPBM has been shown to modulate neural oscillations (Wang et al., 2019; Zomorrodi et al., 2019) [50,51]. Future studies might deeply investigate this point, by studying the potential correlation between improvements in autism severity and EEG changes.

Importantly, the results of this retrospective study suggested that tPBM is safe, since all participants tolerated the stimulations well, even if this technique was previously associated with treatment-emergent side effects, such as headache, strange taste in mouth and decreased appetite (Cassano et al., 2019) [26].

Despite the interesting results, these findings should be evaluated considering some limitations. Results could be partly explained by the placebo effect. Indeed, some elements including regular contact between patients and therapists, and patients‘ expectations to benefit from treatment (in this case parents’ expectations) have been widely reported in the literature as contextual factors that can determine an improvement in symptomatology as the treatment itself (Brody 2018; Kjær et al., 2020) [52,53]. Therefore, further research with well-designed studies, including a double-blind administration of the intervention, and a placebo group, is warranted.

Future studies might use neuroimaging techniques, which could help to understand whether the clinical improvement reported here is associated with functional or maturational changes at the level of a specific network, as has been shown in Alzheimer’s disease, where the improvement in clinical manifestation after tPBM treatment was associated with a reduction in tau and beta-amyloid levels (Chao, 2019) [35]. Furthermore, additional tools that are also able to measure other domains characterizing the ASD might be employed, in order to better understand, for example, whether the tPBM had a better effect on other cognitive domains, such as language.

In conclusion, tPBM represents a promising intervention for children and adolescents with ASD, considering also its practicality and the freedom of movement it offers. If other studies will confirm our findings, tPBM could represent a promising device for moving forward to a more precision medicine approach, on the road to personalized treatment in the realm of neurodevelopmental disorders.

## Figures and Tables

**Figure 1 children-09-00755-f001:**
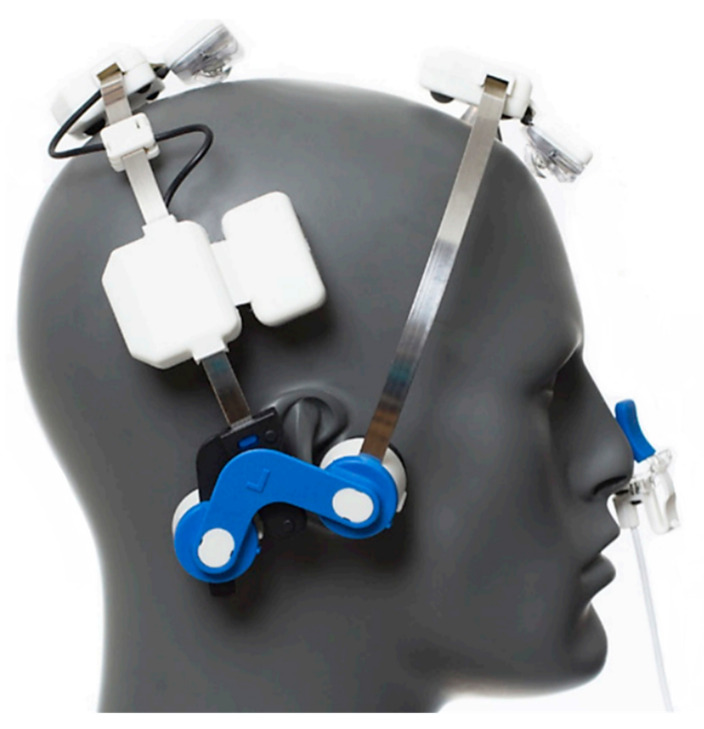
Positioning of the tPBM on the head, which is composed of a helmet and a nasal stimulator.

**Figure 2 children-09-00755-f002:**
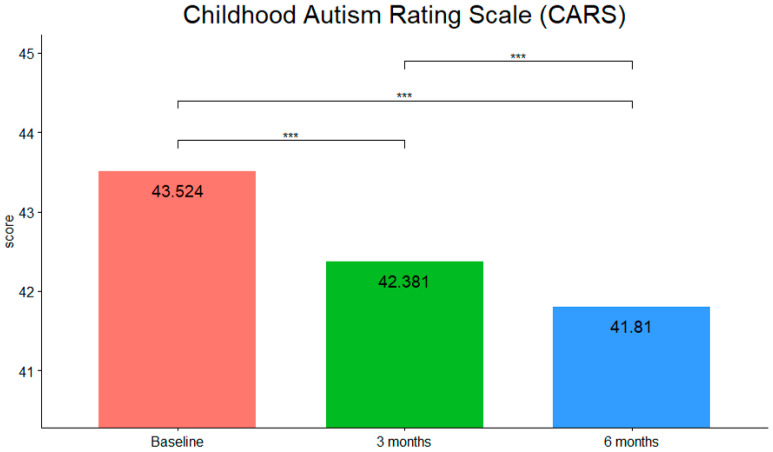
CARS mean scores at the three timepoints (***: *p*-value < 0.001).

**Figure 3 children-09-00755-f003:**
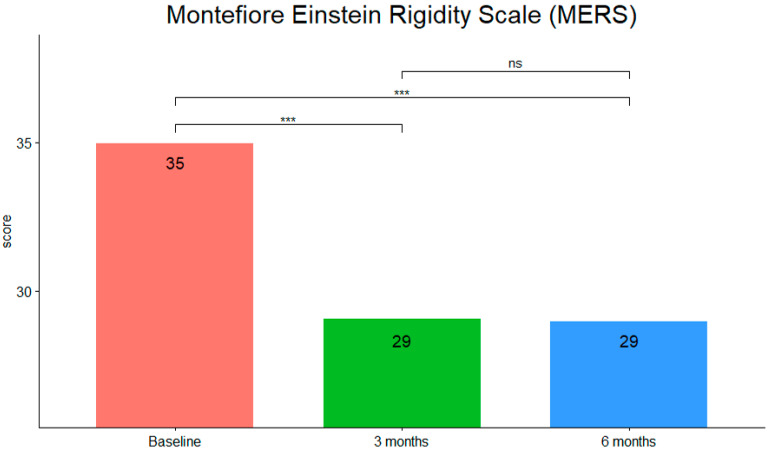
MERS mean scores at the three timepoints (***: *p*-value < 0.001; ns, not significant).

**Figure 4 children-09-00755-f004:**
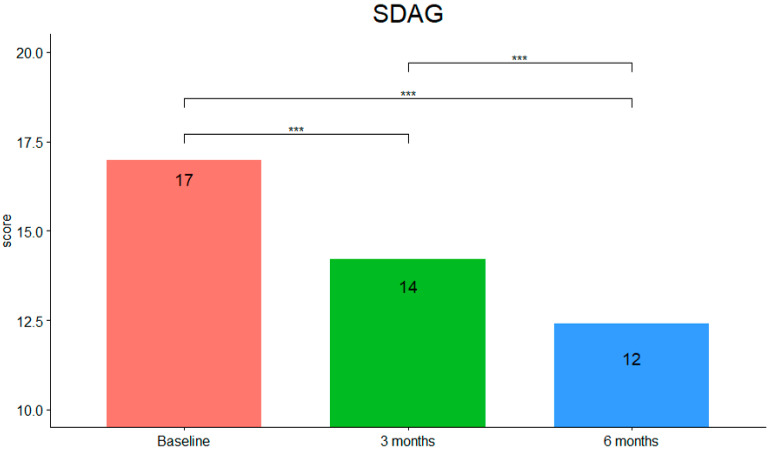
SDAG mean scores at the different timepoints (***: *p*-value < 0.001).

**Figure 5 children-09-00755-f005:**
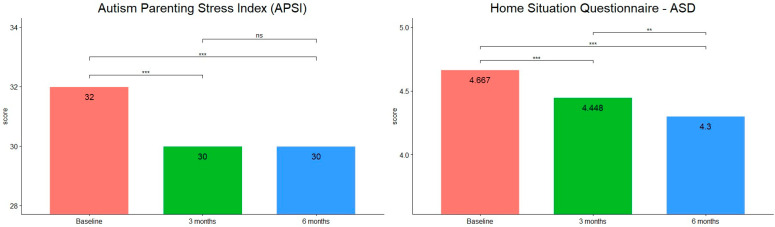
Median scores of HSQ-ASD and ASPI at the three timepoints (***: *p*-value < 0.001; **: *p*-value < 0.01; ns, not significant).

**Figure 6 children-09-00755-f006:**
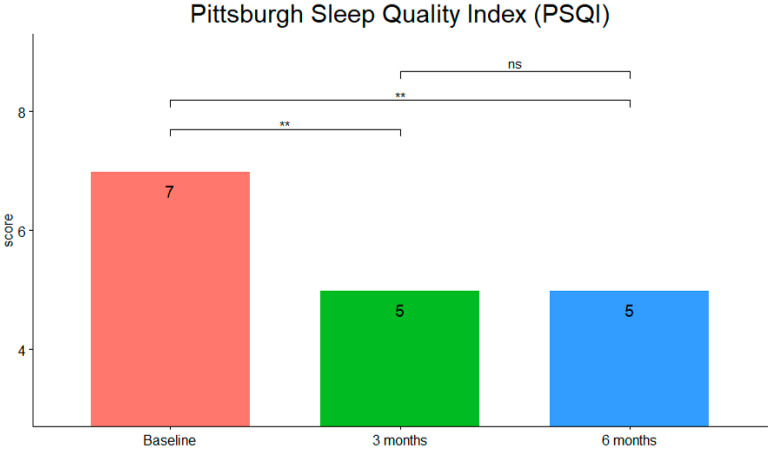
Median scores of PSQI at the three timepoints (**: *p*-value < 0.01; ns, not significant).

**Table 1 children-09-00755-t001:** Demographical data of the patients as well as comorbidities and ongoing treatments (ADHD, Attention Deficit Hyperactivity Disorder; SAD, Social Anxiety Disorder; ODD, Oppositional Defiant Disorder; CBT, Cognitive-Behavioral Therapy).

PATIENT	AGE	GENDER	COMORBIDITIES	MEDICATION	OTHER TREATMENTS
1	9	M	ADHD	Omega-3; Melatonin; Probiotics; Phosphatidylserine	CBT; Speech therapy
2	5	M	ADHD; SAD	Melatonin; Probiotics; Phosphatidylserine	CBT; Parent Training
3	7	F	ODD	Probiotics	CBT; Speech Therapy
4	6	F	SAD	Omega-3	CBT
5	12	M	ADHD	Omega-3; Probiotics; Phosphatidylserine	CBT; Speech Therapy
6	7	F	ADHD	Melatonin; Phosphatidylserine	CBT; Speech Therapy
7	15	M		Omega-3; Melatonin; Probiotics	CBT; Speech Therapy
8	14	M	SAD	Melatonin	CBT; Speech Therapy
9	7	M	ODD		CBT; Speech Therapy; Parent Training
10	7	M	ODD; SAD	Omega-3; Melatonin; Probiotics	
11	8	M		Melatonin	
12	5	F	SAD		CBT; Speech Therapy
13	8	F	ODD	Omega-3	CBT
14	8	M	ADHD	Phosphatidylserine	CBT; Speech Therapy
15	10	M	ADHD	Probiotics; Phosphatidylserine	
16	11	F	ADHD; SAD	Omega-3; Phosphatidylserine	CBT; Speech Therapy
17	7	F	ADHD	Omega-3; Melatonin; Probiotics; Phosphatidylserine	CBT; Speech Therapy
18	7	M		Omega-3; Melatonin	CBT; Speech Therapy; Parent Training
19	14	F	ADHD; SAD	Omega-3; Probiotics; Phosphatidylserine	CBT; Speech Therapy
20	14	M			CBT; Speech Therapy
21	10	M	ADHD	Melatonin; Phosphatidylserine	CBT; Speech Therapy

**Table 2 children-09-00755-t002:** Parameters of the Vielight PBM device (LED, Light-Emitting Diode).

Device Parameter	LED		
	Posterior Transcranial LEDs	Anterior Transcranial LED	Intranasal LED
Power output	100 mW	75 mW	25 mW
Power density	100 mW/cm^2^	75 mW/cm^2^	25 mW/cm^2^
Energy delivered	60 J	45 J	15 J
Energy density per LED	60 J/cm^2^	45 J/cm^2^	15 J/cm^2^

## Data Availability

The data presented in this study are available on request from the corresponding author. The data are not publicly available due to privacy.

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
