# Peer review of "Transcranial Photobiomodulation for the Treatment of Children with Autism Spectrum Disorder (ASD): A Retrospective Study"

_children, 2022, doi:10.3390/children9050755_

Round 1

Reviewer 1 Report

I would like to thank the author for sharing their work with us. This is a very valuable manuscript, with a lot of potential buried in it, and would be well served by a careful revision thus strengthening. Below is a list of suggestions to this end:

  1. In this manuscript, author reports their findings from a prospective (not retrospective), open-label single group design study of tPBM in ASD. The title of the manuscript is therefore misleading. 
  2. Transcranial photobiomodulation is a rather new concept. Therefore, nomenclature is still forming. Currently, the agreed upon term is Transcranial Photobiomodulation, and agreed upon abbreviation is tPBM. Author needs to be clear and remain consistent throughout the manuscript when identifying this intervention. They use several terms interchangeably, including PBM, t-PBM, NIR, LED, TLT, although each refers to something different. NIR is the light, LED is the source of light, and photobiomodulation is the modality of using light (photo-) to induce a change (-modulation) in the target tissue (-bio-). TLT and tPBM are forms of PBM applied to a patient's head, transcranially. 
  3. Lines 44-51: In the Introduction section, Author reviews existing treatment approaches for ASD. However, it needs to be clearly stated that as of now, there exists no treatments that address the core features of ASD, and that current treatment interventions target comorbid disorders (mood. Anxiety disorders, ADHD) or consequences of core features (aggressive behaviors).
  4. Lines 53-56: It is premature to claim whether tPBM has ability to modulate neuroplasticity, or whether it is the modulation that leads to clinical outcome. In fact, investigating whether tPBM has such ability is the purpose of this and future research. More appropriate description of its current state could be that tPBM may have a potential to modulate neuroplasticity.
  5. Line 71: I am not sure what this sentence is aiming to state, so should be revised.
  6. Line 78: final results of this study were published in January 2022, and showed statistically and clinically significant reductions in SRS total score, and social awareness, communication, motivation and RRB subscales. Below, I list the reference for the Author’s consideration:

Ceranoglu TA, Cassano P, Hoskova B, Green A, Dallenbach N, DiSalvo M, Biederman J, Joshi G. Transcranial Photobiomodulation in Adults with High-Functioning Autism Spectrum Disorder: Positive Findings from a Proof-of-Concept Study. Photobiomodul Photomed Laser Surg. 2022 Jan;40(1):4-12. doi: 10.1089/photob.2020.4986. Epub 2021 Dec 23. PMID: 34941429.

  1. Line 88: the cited Salgueiro 2020 manuscript is not a treatment report, but description of a clinical study that will be conducted. Therefore, no side effect information or treatment outcome is available in that manuscript. Should not be referenced in that capacity.
  2. Line 90: it is Faulhaber, not Fauhauer. Also, this is one of the places in this manuscript where NIR, phototherapy and photobiomodulation all get lumped together. Phototherapy is a form of photobiomodulation; however, the wavelength used in phototherapy of neonates for jaundice is usually around 460nm which is in blue range, while tPBM usually employs NIR (800nm or more). I suspect what Author means is, although phototherapy may often be considered side=effect free due to its noninvasive nature, this is incorrect. In fact, phototherapy may pose adverse events that depend on the wavelength of light used.
  3. Line 102-105: Author starts describing the device used in their study, and mentions areas of DMN as targets. Comments about targeted areas and their relevance to ASD core features should be expanded in more detail in Introduction section. Description of Vielight should be placed in Methods section.
    1. During their description of target areas, Author compares their method to Leisman 2018 study, but leave it at saying “not only the temporal lobe…” which then begs the sentence be followed by “…, but also….”
    2. Authors may still describe Vielight here, but they should also describe why they chose Vielight device, why targeted the said areas, and why chose the specific wavelength.
  4. Materals and Methods section: this is not a retrospective study, but seems to be a prospective, open-label, single group design.
  5. Line 116: how were the subjects selected from what pool? That should be described here. The number of subjects enrolled and their demographic data should be described in Results section.
  6. Line 125-135: what areas are targeted in this intervention? This is later described in some detail around line 173, but should be done here.
  7. Line1377-153: it is now customary to include a separate tabledisplaying the parameters of the light applied and device used in tPBM.
  8. Line 149: what ventral brain areas are targeted?
  9. The topographical presentation of the manuscript seems incomplete. For instance, there are elements in Introduction section that should indeed be included in Methods section, and Results are reported in Methods section. 
  10. Line 182: the instruments are described here, but also started in Introduction. Would review all instruments at Methods. If Author need to review a specific instrument, especially the main outcome one, perhaps that can be done in Introduction section, but detailed information should be in Methods section.
  11. Results section: this section starts with description of statistics, which should be done in Methods section, e.g. under a title of Statistical Analyses
  12. a table of demographic data and clinical characteristics should be provided. What were the diagnoses encountered, comorbidities, medication treatment status including what medications, how many different medications on average participants received, other treatment modalities (psychotherapy, occupational therapy, speech/language therapy, etc.)
  13. tables and figures: please include numeric data of each group, a legend in each table, p values, detailed time points (Baseline, 3 months, 6 months).
  14. Discussion section, line 259: It is no more necessary to repeat the results and p values here to qualify the discussion points. Instead, please focus on interpreting the salient findings. Start with interpreting findings from the primary outcome measures, and then secondary outcome measures, and finally the safety and tolerability outcomes.
  15. Line 280: The study published by Ceranoglu, et al., used LED device.
  16. Line 284: Author needs to explicitly state and discuss the basic working principle and expand their point by providing a summary of findings from Bronchetti, et al. study.
  17. Line 315: a rather tortured sentence, please revise.
  18. Not sure whether keywords of "ADHD," and "trauma" relates to this manuscript. Would consider instead, "children and adolescents," or "treatment," or "neuromodulation," or any combination.

Author Response

  • In this manuscript, author reports their findings from a prospective (not retrospective), open-label single group design study of tPBM in ASD. The title of the manuscript is therefore misleading. 

This is a retrospective study; therefore we are sorry whether finding appeared to be reported prospectively. We thank the reviewer for pointing it out. We have modified the presentation of the study, to make it more coherent with a retrospective study.

  • Transcranial photobiomodulation is a rather new concept. Therefore, nomenclature is still forming. Currently, the agreed upon term is Transcranial Photobiomodulation, and agreed upon abbreviation is tPBM. Author needs to be clear and remain consistent throughout the manuscript when identifying this intervention. They use several terms interchangeably, including PBM, t-PBM, NIR, LED, TLT, although each refers to something different. NIR is the light, LED is the source of light, and photobiomodulation is the modality of using light (photo-) to induce a change (-modulation) in the target tissue (-bio-). TLT and tPBM are forms of PBM applied to a patient's head, transcranially. 

Thank you for your comment. We have changed the terminology throughout the text based on your suggestion.

  • Lines 44-51: In the Introduction section, Author reviews existing treatment approaches for ASD. However, it needs to be clearly stated that as of now, there exists no treatments that address the core features of ASD, and that current treatment interventions target comorbid disorders (mood. Anxiety disorders, ADHD) or consequences of core features (aggressive behaviors).

We have added the requested clarification.

  • Lines 53-56: It is premature to claim whether tPBM has ability to modulate neuroplasticity, or whether it is the modulation that leads to clinical outcome. In fact, investigating whether tPBM has such ability is the purpose of this and future research. More appropriate description of its current state could be that tPBM may have a potential to modulate neuroplasticity.

Thank you for your comment, but in lines 53-56 we talk about non-invasive brain stimulation in general and not about the specific case of tPBM.

  • Line 71: I am not sure what this sentence is aiming to state, so should be revised.

Thank you for the suggestion. We have modified the sentence.  

  • Line 78: final results of this study were published in January 2022, and showed statistically and clinically significant reductions in SRS total score, and social awareness, communication, motivation and RRB subscales. Below, I list the reference for the Author’s consideration:

Ceranoglu TA, Cassano P, Hoskova B, Green A, Dallenbach N, DiSalvo M, Biederman J, Joshi G. Transcranial Photobiomodulation in Adults with High-Functioning Autism Spectrum Disorder: Positive Findings from a Proof-of-Concept Study. Photobiomodul Photomed Laser Surg. 2022 Jan;40(1):4-12. doi: 10.1089/photob.2020.4986. Epub 2021 Dec 23. PMID: 34941429.

Thank you for the suggestion. We have updated the reference.

  • Line 88: the cited Salgueiro 2020 manuscript is not a treatment report, but description of a clinical study that will be conducted. Therefore, no side effect information or treatment outcome is available in that manuscript. Should not be referenced in that capacity.

Thank you for pointing it out. There was an error in the reference. It should be the paper by Salgueiro et al., 2021

  • Line 90: it is Faulhaber, not Fauhauer. Also, this is one of the places in this manuscript where NIR, phototherapy and photobiomodulation all get lumped together. Phototherapy is a form of photobiomodulation; however, the wavelength used in phototherapy of neonates for jaundice is usually around 460nm which is in blue range, while tPBM usually employs NIR (800nm or more). I suspect what Author means is, although phototherapy may often be considered side=effect free due to its noninvasive nature, this is incorrect. In fact, phototherapy may pose adverse events that depend on the wavelength of light used.

Thank you for your comment. We have corrected the reference and clarified the sentence based on your comment.

  • Line 102-105: Author starts describing the device used in their study, and mentions areas of DMN as targets. Comments about targeted areas and their relevance to ASD core features should be expanded in more detail in Introduction section. Description of Vielight should be placed in Methods section.

The association between DMN’s alterations and ASD core features was described previously in the introduction (line 40-43).

    • During their description of target areas, Author compares their method to Leisman 2018 study, but leave it at saying “not only the temporal lobe…” which then begs the sentence be followed by “…, but also….”

Thank you for your advice. We have already structured the sentence in a different way but with the same meaning:  tPBM “stimulates the default mode network (DMN) and not only the temporal lobe as it was done in the study by Leisman and collegues”.

    • Authors may still describe Vielight here, but they should also describe why they chose Vielight device, why targeted the said areas, and why chose the specific wavelength.

This information, as far as applicable in our opinion, is already present in the Methods section.

  • Materals and Methods section: this is not a retrospective study, but seems to be a prospective, open-label, single group design.

We are sorry whether description of the study appeared to be reported prospectively. We thank the reviewer for pointing it out. We have modified the presentation of the study, to make it more coherent with a retrospective study, as it is.

  • Line 116: how were the subjects selected from what pool? That should be described here. The number of subjects enrolled and their demographic data should be described in Results section.

Thank you for your comment. A more detailed description was added. Demographic data have been added to the Results section.

  • Line 125-135: what areas are targeted in this intervention? This is later described in some detail around line 173, but should be done here.

Thank you for your comment. We have modified the period based on your advice.

  • Line1377-153: it is now customary to include a separate tabledisplaying the parameters of the light applied and device used in tPBM.

We have added the table as suggested.

  • Line 149: what ventral brain areas are targeted?

Thank you for your comment. We have added this information.

  • The topographical presentation of the manuscript seems incomplete. For instance, there are elements in Introduction section that should indeed be included in Methods section, and Results are reported in Methods section. 

Thank you for your comment. We have modified the organization of information in the various sections of the paper

  • Line 182: the instruments are described here, but also started in Introduction. Would review all instruments at Methods. If Author need to review a specific instrument, especially the main outcome one, perhaps that can be done in Introduction section, but detailed information should be in Methods section.

Thank you. We agree and the text was modified accordingly.

  • Results section: this section starts with description of statistics, which should be done in Methods section, e.g. under a title of Statistical Analyses

Thanks for pointing that out. We have included that description in a "statistical analyses" paragraph

  • a table of demographic data and clinical characteristics should be provided. What were the diagnoses encountered, comorbidities, medication treatment status including what medications, how many different medications on average participants received, other treatment modalities (psychotherapy, occupational therapy, speech/language therapy, etc.)

We have added the table with the missing information.

  • tables and figures: please include numeric data of each group, a legend in each table, p values, detailed time points (Baseline, 3 months, 6 months).

Thank you for your advice. Images have been modified accordingly.

  • Discussion section, line 259: It is no more necessary to repeat the results and p values here to qualify the discussion points. Instead, please focus on interpreting the salient findings. Start with interpreting findings from the primary outcome measures, and then secondary outcome measures, and finally the safety and tolerability outcomes.

Thank for your suggestion. We have removed the unnecessary information

  • Line 280: The study published by Ceranoglu, et al., used LED device.

Thank you. The text has been modified accordingly.

  • Line 284: Author needs to explicitly state and discuss the basic working principle and expand their point by providing a summary of findings from Bronchetti, et al. study.

Thank you for your advice. Although we understand what the reviewer means, we believe that an explanation of the findings from Bronchetti et al., would be beyond the scope of our paper.

  • Line 315: a rather tortured sentence, please revise.

Thank you for your comment. We have modified the sentence.

  • Not sure whether keywords of "ADHD," and "trauma" relates to this manuscript. Would consider instead, "children and adolescents," or "treatment," or "neuromodulation," or any combination.

Actually, the keywords you listed do not correspond to our keywords.  We have listed instead: “Autism Spectrum Disorders; ASD; Photobiomodulation; LED; Near-Infrared; NIR; Neuromodulation”

Reviewer 2 Report

It was my pleasure to review the manuscript: Transcranial Photobiomodulation for the treatment of children with Autism Spectrum Disorder (ASD): a retrospective study. I think the manuscript is valuable to current knowledge of using tPBM in children with autism but requires a few clarifications.

1 Authors used tBPM for home-based treatment.  How much safety at home was implied?

2 The protocol included 5 stimulations per week for 6 months. What was the rationale for this?

3 It was not clear how many points of difference in CARS score from 3 months and 6 months were observed to assess not just efficacy but the significance of treatment.

Author Response

1 Authors used tBPM for home-based treatment.  How much safety at home was implied?

An explanation about safety at home is provided between lines 200-207. Parents were regularly contacted in order to assess possible side effects.

2 The protocol included 5 stimulations per week for 6 months. What was the rationale for this?

Based on scientific literature and on indications by the Vielight company, which produces the device, the recommended used was of 5-6 stimulations per week

3 It was not clear how many points of difference in CARS score from 3 months and 6 months were observed to assess not just efficacy but the significance of treatment.

Thank you for your comment. We have included the mean data for the three timepoints, in order to make clearer the difference in scores, in addition to the already present pairwise analysis.